# The Option Keyboard
# Combining Skills in Reinforcement Learning

**André Barreto, Diana Borsa, Shaobo Hou, Gheorghe Comanici, Eser Aygün,**
**Philippe Hamel**, **Daniel Toyama**, **Jonathan Hunt**, **Shibl Mourad**, **David Silver**, **Doina Precup**

`{andrebarreto,borsa,shaobohou,gcomanici,eser}@google.com`
`{hamelphi,kenjitoyama,jjhunt,shibl,davidsilver,doinap}@google.com`

DeepMind

## Abstract

The ability to combine known skills to create new ones may be crucial in the solution of complex reinforcement learning problems that unfold over extended periods. We argue that a robust way of combining skills is to define and manipulate them in the space of pseudo-rewards (or "cumulants"). Based on this premise, we propose a framework for combining skills using the formalism of options. We show that every deterministic option can be unambiguously represented as a cumulant defined in an extended domain. Building on this insight and on previous results on transfer learning, we show how to approximate options whose cumulants are linear combinations of the cumulants of known options. This means that, once we have learned options associated with a set of cumulants, we can instantaneously synthesise options induced by any linear combination of them, without any learning involved. We describe how this framework provides a hierarchical interface to the environment whose abstract actions correspond to combinations of basic skills. We demonstrate the practical benefits of our approach in a resource management problem and a navigation task involving a quadrupedal simulated robot.

## 1   Introduction

In reinforcement learning (RL) an agent takes actions in an environment in order to maximise the amount of reward received in the long run [25]. This textbook definition of RL treats actions as atomic decisions made by the agent at every time step. Recently, Sutton [23] proposed a new view on action selection. In order to illustrate the potential benefits of his proposal Sutton resorts to the following analogy. Imagine that the interface between agent and environment is a *piano keyboard*, with each key corresponding to a possible action. Conventionally the agent plays one key at a time and each note lasts exactly one unit of time. If we expect our agents to do something akin to playing music, we must generalise this interface in two ways. First, we ought to allow notes to be arbitrarily long—that is, we must replace actions with *skills*. Second, we should be able to also play *chords*.

The argument in favour of temporally-extended courses of actions has repeatedly been made in the literature: in fact, the notion that agents should be able to reason at multiple temporal scales is one of the pillars of hierarchical RL [7, 18, 26, 8, 17]. The insight that the agent should have the ability to *combine* the resulting skills is a far less explored idea. This is the focus of the current work.

The possibility of combining skills replaces a monolithic action set with a combinatorial counterpart: by learning a small set of basic skills ("keys") the agent should be able to perform a potentially very large number of combined skills ("chords"). For example, an agent that can both walk and grasp an object should be able to walk while grasping an object without having to learn a new skill. According

to Sutton [23], this combinatorial action selection process "could be the key to generating behaviour with a good mix of preplanned coherence and sensitivity to the current situation."

But how exactly should one combine skills? One possibility is to combine them in the space of policies: for example, if we look at policies as distribution over actions, a combination of skills can be defined as a mixture of the corresponding distributions. One can also combine parametric policies by manipulating the corresponding parameters. Although these are feasible solutions, they fail to capture possible *intentions* behind the skills. Suppose the agent is able to perform two skills that can be associated with the same objective—distinct ways of grasping an object, say. It is not difficult to see how combinations of the corresponding behaviours can completely fail to accomplish the common goal. We argue that a more robust way of combining skills is to do so directly in the goal space, using pseudo-rewards or *cumulants* [25]. If we associate each skill with a cumulant, we can combine the former by manipulating the latter. This allows us to go beyond the direct prescription of behaviours, working instead in the space of intentions.

Combining skills in the space of cumulants poses two challenges. First, we must establish a well-defined mapping between cumulants and skills. Second, once a combined cumulant is defined, we must be able to perform the associated skill without having to go through the slow process of learning it. We propose to tackle the former by adopting *options* as our formalism to define skills [26]. We show that there is a large subset of the space of options, composed of deterministic options, in which every element can be unambiguously represented as a cumulant defined in an extended domain. Building on this insight, we extend Barreto et al.'s [3, 4] previous results on transfer learning to show how to approximate options whose cumulants are linear combinations of the cumulants of known options. This means that, once the agent has learned options associated with a collection of cumulants, it can instantaneously synthesise options induced by *any* linear combination of them, *without any learning involved*. Thus, by learning a small set of options, the agent instantaneously has at its disposal a potentially enormous number of combined options. Since we are combining cumulants, and not policies, the resulting options will be truly novel, meaning that they cannot, in general, be directly implemented as a simple alternation of their constituents.

We describe how our framework provides a flexible interface with the environment whose abstract actions correspond to combinations of basic skills. As a reference to the motivating analogy described above, we call this interface the *option keyboard*. We discuss the merits of the option keyboard at the conceptual level and demonstrate its practical benefits in two experiments: a resource management problem and a realistic navigation task involving a quadrupedal robot simulated in MuJoCo [30, 21].

## 2 Background

As usual, we assume the interaction between agent and environment can be modelled as a *Markov decision process* (MDP) [19]. An MDP is a tuple $M \equiv (\mathcal{S}, \mathcal{A}, p, r, \gamma)$, where $\mathcal{S}$ and $\mathcal{A}$ are the state and action spaces, $p(\cdot|s, a)$ gives the next-state distribution upon taking action $a$ in $s$, $r : \mathcal{S} \times \mathcal{A} \times \mathcal{S} \mapsto \mathbb{R}$ specifies the reward associated with the transition $s \xrightarrow{a} s'$, and $\gamma \in [0, 1)$ is the discount factor.

The objective of the agent is to find a *policy* $\pi : \mathcal{S} \mapsto \mathcal{A}$ that maximises the expected *return* $G_t \equiv \sum_{i=0}^{\infty} \gamma^i R_{t+i}$, where $R_t = r(S_t, A_t, S_{t+1})$. A principled way to address this problem is to use methods derived from dynamic programming, which usually compute the *action-value function* of a policy $\pi$ as: $Q^\pi(s, a) \equiv \mathbb{E}^\pi [G_t | S_t = s, A_t = a]$, where $\mathbb{E}^\pi[\cdot]$ denotes expectation over the transitions induced by $\pi$ [19]. The computation of $Q^\pi(s, a)$ is called *policy evaluation*. Once $\pi$ has been evaluated, we can compute a greedy policy

$$\pi'(s) \in \operatorname{argmax}_a Q^\pi(s, a) \text{ for all } s \in \mathcal{S}. \tag{1}$$

It can be shown that $Q^{\pi'}(s, a) \geq Q^\pi(s, a)$ for all $(s, a) \in \mathcal{S} \times \mathcal{A}$, and hence the computation of $\pi'$ is referred to as *policy improvement*. The alternation between policy evaluation and policy improvement is at the core of many RL algorithms, which usually carry out these steps approximately. Here we will use a tilde over a symbol to indicate that the associated quantity is an approximation (*e.g.*, $\tilde{Q}^\pi \approx Q^\pi$).

### 2.1 Generalising policy evaluation and policy improvement

Following Sutton and Barto [25], we call any signal defined as $c : \mathcal{S} \times \mathcal{A} \times \mathcal{S} \mapsto \mathbb{R}$ a *cumulant*. Analogously to the conventional value function $Q^\pi$, we define $Q_c^\pi$ as the expected discounted sum of

cumulant $c$ under policy $\pi$ [27]. Given a policy $\pi$ and a set of cumulants $\mathcal{C}$, we call the evaluation of $\pi$ under all $c \in \mathcal{C}$ *generalised policy evaluation* (GPE) [2]. Barreto et al. [3, 4] propose an efficient form of GPE based on *successor features*: they show that, given cumulants $c_1, c_2, ..., c_d$, for any $c = \sum_i w_i c_i$, with $\boldsymbol{w} \in \mathbb{R}^d$,

$$Q_c^\pi(s, a) \equiv \mathbb{E}^\pi \left[ \sum_{k=0}^\infty \gamma^k \sum_{i=1}^d w_i C_{i,t+k} | S_t = s, A_t = a \right] = \sum_{i=1}^d w_i Q_{c_i}^\pi(s, a), \qquad (2)$$

where $C_{i,t} \equiv c_i(S_t, A_t, R_t)$. Thus, once we have computed $Q_{c_1}^\pi, Q_{c_1}^\pi, ..., Q_{c_d}^\pi$, we can instantaneously evaluate $\pi$ under any cumulant in the set $\mathcal{C} \equiv \{ c = \sum_i w_i c_i \, | \, \boldsymbol{w} \in \mathbb{R}^d \}$.

Policy improvement can also be generalised. In Barreto et al.'s [3] *generalised policy improvement* (GPI) the improved policy is computed based on a set of value functions. Let $Q_c^{\pi_1}, Q_c^{\pi_2}, ...Q_c^{\pi_n}$ be the action-value functions of $n$ policies $\pi_i$ under cumulant $c$, and let $Q_c^{\max}(s, a) = \max_i Q_c^{\pi_i}(s, a)$ for all $(s, a) \in \mathcal{S} \times \mathcal{A}$. If we define

$$\pi(s) \in \operatorname{argmax}_a Q_c^{\max}(s, a) \text{ for all } s \in \mathcal{S}, \qquad (3)$$

then $Q_c^\pi(s, a) \geq Q_c^{\max}(s, a)$ for all $(s, a) \in \mathcal{S} \times \mathcal{A}$. This is a strict generalisation of standard policy improvement (1). The guarantee extends to the case in which GPI uses approximations $\tilde{Q}_c^{\pi_i}$ [3].

## 2.2 Temporal abstraction via options

As discussed in the introduction, one way to get temporal abstraction is through the concept of *options* [26]. Options are temporally-extended courses of actions. In their more general formulation, options can depend on the entire *history* between the time $t$ when they were initiated and the current time step $t + k$, $h_{t:t+k} \equiv s_t a_t s_{t+1}...a_{t+k-1} s_{t+k}$. Let $\mathcal{H}$ be the space of all possible histories; a *semi-Markov option* is a tuple $o \equiv (\mathcal{I}_o, \pi_o, \beta_o)$ where $\mathcal{I}_o \subset \mathcal{S}$ is the set of states where the option can be initiated, $\pi_o : \mathcal{H} \mapsto \mathcal{A}$ is a policy over histories, and $\beta_o : \mathcal{H} \mapsto [0, 1]$ gives the probability that the option terminates after history $h$ has been observed [26]. It is worth emphasising that semi-Markov options depend on the history since their initiation, but not before.

# 3 Combining options

In the previous section we discussed how several key concepts in RL can be generalised: rewards with cumulants, policy evaluation with GPE, policy improvement with GPI, and actions with options. In this section we discuss how these concepts can be used to combine skills.

## 3.1 The relation between options and cumulants

We start by showing that there is a subset of the space of options in which every option can be unequivocally represented as a cumulant defined in an extended domain.

First we look at the relation between policies and cumulants. Given an MDP $(\mathcal{S}, \mathcal{A}, p, \cdot, \gamma)$, we say that a cumulant $c_\pi : \mathcal{S} \times \mathcal{A} \times \mathcal{S} \mapsto \mathbb{R}$ *induces* a policy $\pi : \mathcal{S} \mapsto \mathcal{A}$ if $\pi$ is optimal for the MDP $(\mathcal{S}, \mathcal{A}, p, c_\pi, \gamma)$. We can always define a cumulant $c_\pi$ that induces a given policy $\pi$. For instance, if we make

$$c_\pi(s, a, \cdot) = \begin{cases} 0 \text{ if } a = \pi(s); \\ z \text{ otherwise,} \end{cases} \qquad (4)$$

where $z < 0$, it is clear that $\pi$ is the only policy that achieves the maximum possible value $Q^\pi(s, a) = Q^*(s, a) = 0$ on all $(s, a) \in \mathcal{S} \times \mathcal{A}$. In general, the relation between policies and cumulants is a many-to-many mapping. First, there is more than one cumulant that induces the same policy: for example, any $z < 0$ in (4) will clearly lead to the same policy $\pi$. There is thus an infinite set of cumulants $\mathcal{C}_\pi$ associated with $\pi$. Conversely, although this is not the case in (4), the same cumulant can give rise to multiple policies if more than one action achieves the maximum in (1).

Given the above, we can use any cumulant $c_\pi \in \mathcal{C}_\pi$ to refer to policy $\pi$. In order to extend this possibility to options $o = (\mathcal{I}_o, \pi_o, \beta_o)$ we need two things. First, we must define cumulants in the space of histories $\mathcal{H}$. This will allow us to induce semi-Markov policies $\pi_o : \mathcal{H} \mapsto \mathcal{A}$ in a way that is analogous to (4). Second, we need cumulants that also induce the initiation set $\mathcal{I}_o$ and the termination function $\beta_o$. We propose to accomplish this by augmenting the action space.

Let $\tau$ be a *termination action* that terminates option $o$ much like the termination function $\beta_o$. We can think of $\tau$ as a fictitious action and model it by defining an augmented action space $\mathcal{A}^+ \equiv \mathcal{A} \cup \{\tau\}$. When the agent is executing an option $o$, selecting action $\tau$ immediately terminates it. We now show that if we extend the definition of cumulants to also include $\tau$ we can have the resulting cumulant induce not only the option's policy but also its initiation set and termination function. Let $e : \mathcal{H} \times \mathcal{A}^+ \times \mathcal{S} \mapsto \mathbb{R}$ be an *extended cumulant*. Since $e$ is defined over the augmented action space, for each $h \in \mathcal{H}$ we now have a *termination bonus* $e(h, \tau, s) = e(h, \tau)$ that determines the value of interrupting option $o$ after having observed $h$. The extended cumulant $e$ induces an *augmented policy* $\omega_e : \mathcal{H} \mapsto \mathcal{A}^+$ in the same sense that a standard cumulant induces a policy (that is, $\omega_e$ is an optimal policy for the derived MDP whose state space is $\mathcal{H}$ and the action space is $\mathcal{A}^+$; see Appendix A for details). We argue that $\omega_e$ is equivalent to an option $o_e \equiv (\mathcal{I}_e, \pi_e, \beta_e)$ whose components are defined as follows. The policy $\pi_e : \mathcal{H} \mapsto \mathcal{A}$ coincides with $\omega_e$ whenever the latter selects an action in $\mathcal{A}$. The termination function is given by

$$\beta_e(h) = \begin{cases} 1 & \text{if } e(h, \tau) > \max_{a \neq \tau} Q_e^{\omega_e}(h, a), \\ 0 & \text{otherwise.} \end{cases} \tag{5}$$

In words, the agent will terminate after $h$ if the instantaneous termination bonus $e(h, \tau)$ is larger than the maximum expected discounted sum of cumulant $e$ under policy $\omega_e$. Note that when $h$ is a single state $s$, no concrete action has been executed by the option yet, hence it terminates with $\tau$ immediately after its initiation. This is precisely the definition of the initialisation set $\mathcal{I}_e \equiv \{s \,|\, \beta_e(s) = 0\}$.

Termination functions like (5) are always deterministic. This means that extended cumulants $e$ can only represent options $o_e$ in which $\beta_e$ is a mapping $\mathcal{H} \mapsto \{0, 1\}$. In fact, it is possible to show that all options of this type, which we will call *deterministic options*, are representable as an extended cumulant $e$, as formalised in the following proposition (proof in Appendix A):

**Proposition 1.** *Every extended cumulant induces at least one deterministic option, and every deterministic option can be unambiguously induced by an infinite number of extended cumulants.*

## 3.2   Synthesising options using GPE and GPI

In the previous section we looked at the relation between extended cumulants and deterministic options; we now build on this connection to use GPE and GPI to combine options.

Let $\mathcal{E} \equiv \{e_1, e_2, ..., e_d\}$ be a set of extended cumulants. We know that $e_i : \mathcal{H} \times \mathcal{A}^+ \times \mathcal{S} \mapsto \mathbb{R}$ is associated with deterministic option $o_{e_i} \equiv \omega_{e_i}$. As with any other cumulant, the extended cumulants $e_i$ can be linearly combined; it then follows that, for any $\boldsymbol{w} \in \mathbb{R}^d$, $e = \sum_i w_i e_i$ defines a new deterministic option $o_e \equiv \omega_e$. Interestingly, the termination function of $o_e$ has the form (5) with termination bonuses defined as $e(h, \tau) = \sum_i w_i e_i(h, \tau)$. This means that the combined option $o_e$ "inherits" its termination function from its constituents $o_{e_i}$. Since any $\boldsymbol{w} \in \mathbb{R}^d$ defines an option $o_e$, the set $\mathcal{E}$ can give rise to a very large number of combined options.

The problem is of course that for each $\boldsymbol{w} \in \mathbb{R}^d$ we have to actually compute the resulting option $\omega_e$. This is where GPE and GPI come to the rescue. Suppose we have the values of options $\omega_{e_i}$ under all the cumulants $e_1, e_2, ..., e_d$. With this information, and analogously to (2), we can use the fast form of GPE provided by successor features to compute the value of $\omega_{e_j}$ with respect to $e$:

$$Q_e^{\omega_{e_j}}(h, a) = \sum_i w_i Q_{e_i}^{\omega_{e_j}}(h, a). \tag{6}$$

Now that we have all the options $\omega_{e_j}$ evaluated under $e$, we can merge them to generate a new option that does at least as well as, and in general better than, all of them. This is done by applying GPI over the value functions $Q_e^{\omega_{e_j}}$:

$$\tilde{\omega}_e(h) \in \operatorname{argmax}_{a \in \mathcal{A}^+} \max_j Q_e^{\omega_{e_j}}(h, a). \tag{7}$$

From previous theoretical results we know that $\max_j Q_e^{\omega_{e_j}}(h, a) \leq Q_e^{\tilde{\omega}_e}(h, a) \leq Q_e^{\omega_e}(h, a)$ for all $(h, a) \in \mathcal{H} \times \mathcal{A}^+$ [3]. In words, this means that, even though the GPI option $\tilde{\omega}_e$ is not necessarily optimal, following it will in general result in a higher return in terms of cumulant $e$ than if the agent were to execute any of the known options $\omega_{e_j}$. Thus, we can use $\tilde{\omega}_e$ as an approximation to $\omega_e$ *that requires no additional learning*. It is worth mentioning that the action selected by the combined option in (7), $\tilde{\omega}_e(h)$, can be different from $\omega_{e_i}(h)$ for all $i$—that is, the resulting policy cannot, in

general, be implemented as an alternation of its constituents. This highlights the fact that combining cumulants is not the same as defining a higher-level policy over the associated options.

In summary, given a set of cumulants $\mathcal{E}$, we can combine them by picking weights $\boldsymbol{w}$ and computing the resulting cumulant $e = \sum_i w_i e_i$. This can be interpreted as determining how desirable or undesirable each cumulant is. Going back to the example in the introduction, suppose that $e_1$ is associated with walking and $e_2$ is associated with grasping an object. Then, cumulant $e_1 + e_2$ will reinforce both behaviours, and will be particularly rewarding when they are executed together. In contrast, cumulant $e_1 - e_2$ will induce an option that avoids grasping objects, favouring the walking behaviour in isolation and even possibly inhibiting it. Since the resulting option aims at maximising a combination of the cumulants $e_i$, it can itself be seen as a combination of the options $o_{e_i}$.

## 4   Learning with combined options

Given a set of extended cumulants $\mathcal{E}$, in order to be able to combine the associated options using GPE and GPI one only needs the value functions $\mathcal{Q}_{\mathcal{E}} \equiv \{\tilde{Q}_{e_j}^{\omega_{e_i}} \mid \forall (i,j) \in \{1,2,...,d\}^2\}$. The set $\mathcal{Q}_{\mathcal{E}}$ can be constructed using standard RL methods; for an illustration of how to do it with $Q$-learning see Algorithm 3 in App. B.

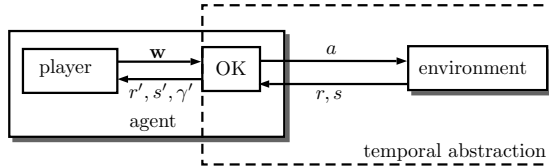

As discussed, once $\mathcal{Q}_{\mathcal{E}}$ has been computed we can use GPE and GPI to synthesise options on the fly. In this case the newly-generated options are fully determined by the vector of weights $\boldsymbol{w} \in \mathbb{R}^d$. Conceptually, we can think of this process as an

Figure 1: OK mediates the interaction between player and environment. The exchange of information between OK and the environment happens at every time step. The interaction between player and OK only happens "inside" the agent when the termination action $\tau$ is selected by GPE and GPI (see Algorithms 1 and 2).

interface between an RL algorithm and the environment: the algorithm selects a vector $\boldsymbol{w}$, hands it over to GPE and GPI, and "waits" until the action returned by (7) is the termination action $\tau$. Once $\tau$ has been selected, the algorithm picks a new $\boldsymbol{w}$, and so on. The RL method is thus interacting with the environment at a higher level of abstraction in which actions are combined skills defined by the vectors of weights $\boldsymbol{w}$. Returning to the analogy with a piano keyboard described in the introduction, we can think of each option $\omega_{e_i}$ as a "key" that can be activated by an instantiation of $\boldsymbol{w}$ whose only non-zero entry is $w_i > 0$. Combined options associated with more general instantiations of $\boldsymbol{w}$ would correspond to "chords". We will thus call the layer of temporal abstraction between algorithm and environment the *option keyboard* (OK). We will also generically refer to the RL method interacting with OK as the "player". Figure 1 shows how an RL agent can be broken into a player and an OK.

Algorithm 1 shows a generic implementation of OK. Given a set of value functions $\mathcal{Q}_{\mathcal{E}}$ and a vector of weights $\boldsymbol{w}$, OK will execute the actions selected by GPE and GPI until the termination action is picked or a terminal state is reached. During this process OK keeps track of the discounted reward accumulated in the interaction with the environment (line 6), which will be returned to the player when the interaction terminates (line 10). As the options $\omega_{e_i}$ may depend on the entire trajectory since their initiation, OK uses an update function $u(h, a, s')$ that retains the parts of the history that are actually relevant for decision making (line 8). For example, if OK is based on Markov options only, one can simply use the update function $u(h, a, s') = s'$.

The set $\mathcal{Q}_{\mathcal{E}}$ defines a specific instantiation of OK; once an OK is in place any conventional RL method can interact with it as if it were the environment. As an illustration, Algorithm 2 shows how a keyboard player that uses a finite set of combined options $\mathcal{W} \equiv \{\boldsymbol{w}_1, \boldsymbol{w}_2, ..., \boldsymbol{w}_n\}$ can be implemented using standard $Q$-learning by simply replacing the environment with OK. It is worth pointing out that if we substitute any other set of weight vectors $\mathcal{W}'$ for $\mathcal{W}$ we can still use the same OK, without the need to relearn the value functions in $\mathcal{Q}_{\mathcal{E}}$. We can even use sets of abstract actions $\mathcal{W}$ that are infinite—as long as the OK player can deal with continuous action spaces [33, 24, 22].

Although the clear separation between OK and its player is instructive, in practice the boundary between the two may be more blurry. For example, if the player is allowed to intervene in all interactions between OK and environment, one can implement useful strategies like option interruption [26]. Finally, note that although we have been treating the construction of OK (Algorithm 3) and its use

(Algorithms 1 and 2) as events that do not overlap in time, nothing keeps us from carrying out the two procedures in parallel, like in similar methods in the literature [1, 32].

---

**Algorithm 1** Option Keyboard (OK)

$$\textbf{Require:} \begin{cases} s \in \mathcal{S} & \text{current state} \\ \boldsymbol{w} \in \mathbb{R}^d & \text{vector of weights} \\ \mathcal{Q}_{\mathcal{E}} & \text{value functions} \\ \gamma \in [0,1) & \text{discount rate} \end{cases}$$

1: $h \leftarrow s; \quad r' \leftarrow 0; \quad \gamma' \leftarrow 1$
2: **repeat**
3:     $a \leftarrow \operatorname{argmax}_{a'} \max_i [\sum_j w_j \tilde{Q}_{e_j}^{\omega_{e_i}}(h, a')]$
4:     **if** $a \neq \tau$ **then**
5:         execute action $a$ and observe $r$ and $s'$
6:         $r' \leftarrow r' + \gamma' r$
7:         **if** $s'$ is terminal $\gamma' \leftarrow 0$ **else** $\gamma' \leftarrow \gamma' \gamma$
8:         $h \leftarrow u(h, a, s')$
9: **until** $a = \tau$ **or** $s'$ is terminal
10: **return** $s', r', \gamma'$

---

**Algorithm 2** $Q$-learning keyboard player

$$\textbf{Require:} \begin{cases} \text{OK} & \text{option keyboard} \\ \mathcal{W} & \text{combined options} \\ \mathcal{Q}_{\mathcal{E}} & \text{value functions} \\ \alpha, \epsilon, \gamma \in \mathbb{R} & \text{hyper-parameters} \end{cases}$$

1: create $\tilde{Q}(s, \boldsymbol{w})$ parametrised by $\boldsymbol{\theta}_Q$
2: select initial state $s \in \mathcal{S}$
3: **repeat forever**
4:     **if** Bernoulli($\epsilon$)=1 **then** $\boldsymbol{w} \leftarrow \text{Uniform}(\mathcal{W})$
5:     **else** $\boldsymbol{w} \leftarrow \operatorname{argmax}_{\boldsymbol{w}' \in \mathcal{W}} \tilde{Q}(s, \boldsymbol{w}')$
6:     $(s', r', \gamma') \leftarrow \text{OK}(s, \boldsymbol{w}, \mathcal{Q}_{\mathcal{E}}, \gamma)$
7:     $\delta \leftarrow r' + \gamma' \max_{\boldsymbol{w}'} \tilde{Q}(s', \boldsymbol{w}') - \tilde{Q}(s, \boldsymbol{w})$
8:     $\boldsymbol{\theta}_Q \leftarrow \boldsymbol{\theta}_Q + \alpha \delta \nabla_{\boldsymbol{\theta}_Q} \tilde{Q}(s, \boldsymbol{w})$   // update $\tilde{Q}$
9:     **if** $s'$ is terminal **then** select initial $s \in \mathcal{S}$
10:     **else** $s \leftarrow s'$

---

## 5 Experiments

We now present our experimental results illustrating the benefits of OK in practice. Additional details, along with further results and analysis, can be found in Appendix C.

### 5.1 Foraging world

The goal in the foraging world is to manage a set of resources by navigating in a grid world and picking up items containing the resources in different proportions. For illustrative purposes we will consider that the resources are nutrients and the items are food. The agent's challenge is to stay healthy by keeping its nutrients within certain bounds. The agent navigates in the grid world using the four usual actions: up, down, left, and right. Upon collecting a food item the agent's nutrients are increased according to the type of food ingested. Importantly, the quantity of each nutrient decreases by a fixed amount at every step, so the desirability of different types of food changes even if no food is consumed. Observations are images representing the configuration of the grid plus a vector indicating how much of each nutrient the agent currently has (see Appendix C.1 for a technical description).

What makes the foraging world particularly challenging is the fact that the agent has to *travel* towards the items to pick them up, adding a spatial aspect to an already complex management problem. The dual nature of the problem also makes it potentially amenable to be tackled with options, since we can design skills that seek specific nutrients and then treat the problem as a management task in which actions are preferences over nutrients. However, the number of options needed can increase exponentially fast. If at any given moment the agent wants, does not want, or does not care about each nutrient, we need $3^m$ options to cover the entire space of preferences, where $m$ is the number of nutrients. This is a typical situation where being able to combine skills can be invaluable.

As an illustration, in our experiments we used $m = 2$ nutrients and 3 types of food. We defined a cumulant $e_i \in \mathcal{E}$ associated with each nutrient as follows: $e_i(h, a, s) = 0$ until a food item is consumed, when it becomes the increase in the associated nutrient. After a food item is consumed we have that $e_i(h, a, s) = -\mathbf{1}\{a \neq \tau\}$, where $\mathbf{1}\{\cdot\}$ is the indicator function—this forces the induced option to terminate, and also illustrates how the definition of cumulants over histories $h$ can be useful (since single states would not be enough to determine whether the agent has consumed a food item). We used Algorithm 3 in Appendix B to compute the 4 value functions in $\mathcal{Q}_{\mathcal{E}}$. We then defined a 8-dimensional abstract action space covering the space of preferences, $\mathcal{W} \equiv \{-1, 0, 1\}^2 - \{[0, 0]\}$, and used it with the $Q$-learning player in Algorithm 2. We also consider $Q$-learning using only the 2 options maximizing each nutrient and a "flat" $Q$-learning agent that does not use options at all.

By modifying the target range of each nutrient we can create distinct scenarios with very different dynamics. Figure 2 shows results in two such scenarios. Note how the relative performance of the two baselines changes dramatically from one scenario to the other, illustrating how the usefulness of options is highly context-dependent. Importantly, as shown by the results of the OK player, the

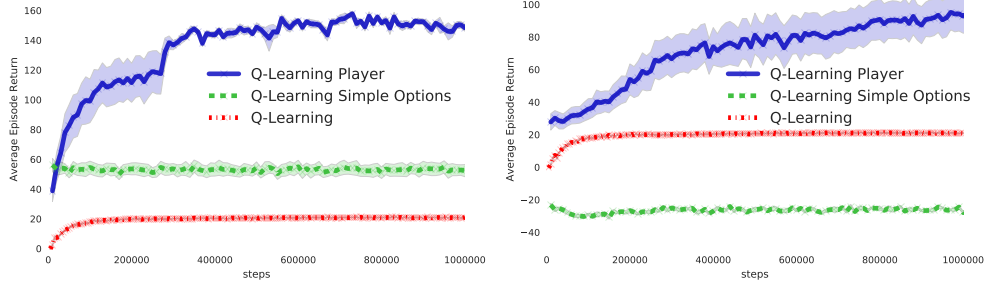

Figure 2: Results on the foraging world. The two plots correspond to different configurations of the environment (see Appendix C.1). Shaded regions are one standard deviation over 10 runs.

ability to combine options in cumulant space makes it possible to synthesise useful behaviour from a given set of options even when they are not useful in isolation.

## 5.2 Moving-target arena

As the name suggests, in the moving-target arena the goal is to get to a target region whose location changes every time the agent reaches it. The arena is implemented as a square room with realistic dynamics defined in the MuJoCo physics engine [30]. The agent is a quadrupedal simulated robot with 8 actuated degrees of freedom; actions are thus vectors in $[-1, 1]^8$ indicating the torque applied to each joint [21]. Observations are 29-dimensional vectors with spatial and proprioception information (Appendix C.2). The reward is always 0 except when the agent reaches the target, when it is 1.

We defined cumulants in order to encourage the agent's displacement in certain directions. Let $\boldsymbol{v}(h)$ be the vector of $(x, y)$ velocities of the agent after observing history $h$ (the velocity is part of the observation). Then, if we want the agent to travel at a certain direction $\boldsymbol{w}$ for $k$ steps, we can define:

$$e_{\boldsymbol{w}}(h, a, \cdot) = \left\{ \begin{array}{l} \boldsymbol{w}^\top \boldsymbol{v}(h) \text{ if length}(h) \leq k; \\ -\mathbf{1}\{a \neq \tau\} \text{ otherwise.} \end{array} \right. \tag{8}$$

The induced option will terminate after $k = 8$ steps as a negative reward is incurred for all histories of length greater than $k$ and actions other than $\tau$. It turns out that even if a larger number of directions $\boldsymbol{w}$ is to be learned, we only need to compute two value functions for each cumulant $e_{\boldsymbol{w}}$. Since for all $e_{\boldsymbol{w}} \in \mathcal{E}$ we have that $e_{\boldsymbol{w}} = w_1 e_{\boldsymbol{x}} + w_2 e_{\boldsymbol{y}}$, where $\boldsymbol{x} = [1, 0]$ and $\boldsymbol{y} = [0, 1]$, we can use (2) to decompose the value function of any option $\omega$ as $Q_{e_{\boldsymbol{w}}}^\omega(h, a) = w_1 Q_{e_{\boldsymbol{x}}}^\omega(h, a) + w_2 Q_{e_{\boldsymbol{y}}}^\omega(h, a)$. Hence, $|\mathcal{Q}_\mathcal{E}| = 2|\mathcal{E}|$, resulting in a 2-dimensional space $\mathcal{W}$ in which $\boldsymbol{w} \in \mathbb{R}^2$ indicates the intended direction of locomotion. Thus, by learning a few options that move along specific directions, the agent is potentially able to synthesise options that travel in *any* direction.

For our experiments, we defined cumulants $e_{\boldsymbol{w}}$ corresponding to the directions $0^o$, $120^o$, and $240^o$. To compute the set of value functions $\mathcal{Q}_\mathcal{E}$ we used Algorithm 3 with $Q$-learning replaced by the deterministic policy gradient (DPG) algorithm [22]. We then used the resulting OK with both discrete and continuous abstract-action spaces $\mathcal{W}$. For finite $\mathcal{W}$ we adopted a $Q$-learning player (Algorithm 2); in this case the abstract actions $\boldsymbol{w}_i$ correspond to $n \in \{4, 6, 8\}$ directions evenly-spaced in the unit circle. For continuous $\mathcal{W}$ we used a DPG player. We compare OK's results with that of DPG applied directly in the original action space and also with $Q$-learning using only the three basic options.

Figure 3 shows our results on the moving-target arena. As one can see by DPG's results, solving the problem in the original action space is difficult because the occurrence of non-zero rewards may depend on a long sequence of lucky actions. When we replace actions with options we see a clear speed up in learning, even if we take into account the training of the options. If in addition we allow for combined options, we observe a significant boost in performance, as shown by the OK players' results. Here we see the expected trend: as we increase $|\mathcal{W}|$ the OK player takes longer to learn but achieves better final performance, as larger numbers of directional options allow for finer control.

These results clearly illustrate the benefits of being able to combine skills, but how much is the agent actually using this ability? In Figure 3 we show a histogram indicating how often combined options are used by OK to implement directions $\boldsymbol{w} \in \mathbb{R}^2$ across the state space (details in App. C.2). As shown, for abstract actions $\boldsymbol{w}$ close to $0^o$, $120^o$ and $240^o$ the agent relies mostly on the 3 options trained to navigate along these directions, but as the intended direction of locomotion gets farther from

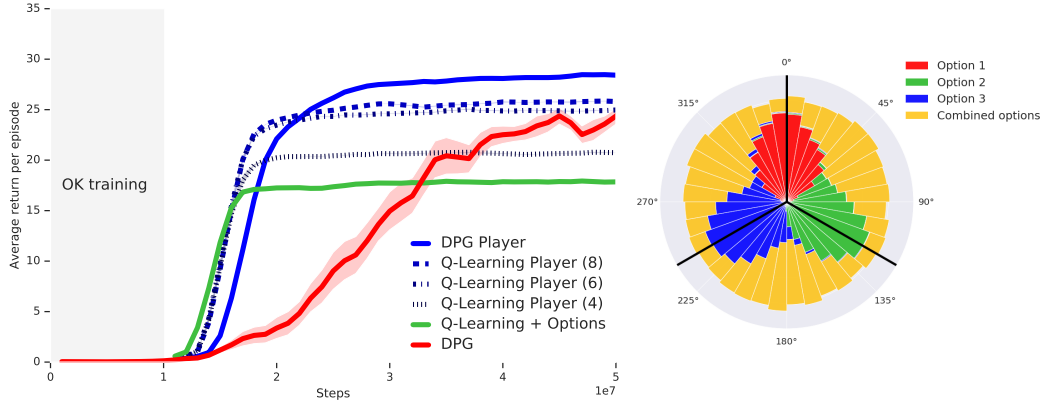

Figure 3: **Left**: Results on the moving-target arena. All players used the *same* keyboard, so they share the same OK training phase. Shaded regions are one standard deviation over 10 runs. **Right**: Histogram of options used by OK to implement directions $w$. Black lines are the three basic options.

these reference points combined options become crucial. This shows how the ability to combine skills can extend the range of behaviours available to an agent without the need for additional learning.[1]

Even if one accepts the premise of this paper that skills should be combined in the space of cumulants, it is natural to ask whether other strategies could be used instead of GPE and GPI. Although we are not aware of any other algorithm that explicitly attempts to combine skills in the space of cumulants, there are methods that do so in the space of value functions [29, 6, 13, 16]. Haarnoja et al. [13] propose a way of combining skills based on entropy-regularised value functions. Given a set of cumulants $e_1, e_2, ..., e_d$, they propose to compute a skill associated with $e = \sum_i w_i e_i$ as follows: $\hat{\omega}_e(h) \in \text{argmax}_{a \in \mathcal{A}^+} \sum_j w_j \hat{Q}_{e_j}^{\omega_{e_j}}(h, a)$, where $\hat{Q}_{e_j}^{\omega_{e_j}}(h, a)$ are entropy-regularised value functions and $w_j \in [-1, 1]$. We will refer to this method as *additive value composition* (AVC).

How well does AVC perform as compared to GPE and GPI? In order to answer this question we reran the previous experiments but now using $\hat{\omega}_e(h)$ as defined above instead of the option $\tilde{\omega}_e(h)$ computed through (6) and (7). In order to adhere more closely to the assumptions underlying AVC, we also repeated the experiment using an entropy-regularised OK [14] (App. C.2). Figure 4 shows the results. As indicated in the figure, GPE and GPI outperform AVC both with the standard and the entropy-regularised OK. A possible explanation for this is given in the accompanying polar scatter chart in Figure 4, which illustrates how much progress each method makes, over the state space, in all directions $w$ (App. C.2). The plot suggests that, in this domain, the directional options implemented through GPI and GPE are more effective in navigating along the desired directions (also see [16]).

# 6 Related work

Previous work has used GPI and successor features, the linear form of GPE considered here, in the context of transfer [3, 4, 5]. A crucial assumption underlying these works is that the reward can be well approximated as $r(s, a, s') \approx \sum_i w_i c_i(s, a, s')$. By solving a regression problem, the agent finds a $w \in \mathbb{R}^d$ that leads to a good approximation of $r(s, a, s')$ and uses it to apply GPE and GPI (equations (2) and (3), respectively). In terms of the current work, this is equivalent to having a keyboard player that is only allowed to play one endless "chord". Through the introduction of a termination action, in this work we replace policies with options that may eventually halt. Since policies are options that never terminate, the previous framework is a special case of OK. Unlike in the previous framework, with OK we can also *chain* a sequence of options, resulting in more flexible behaviour. Importantly, this allows us to completely remove the linearity assumption on the rewards.

We now turn our attention to previous attempts to combine skills with no additional learning. As discussed, one way to do so is to work directly in the space of policies. Many policy-based methods first learn a parametric representation of a lower-level policy, $\pi(\cdot \,|\, s; \theta)$, and then use $\theta \in \mathbb{R}^d$ as the actions for a higher-level policy $\mu : \mathcal{S} \mapsto \mathbb{R}^d$ [15, 10, 12]. One of the central arguments of this paper

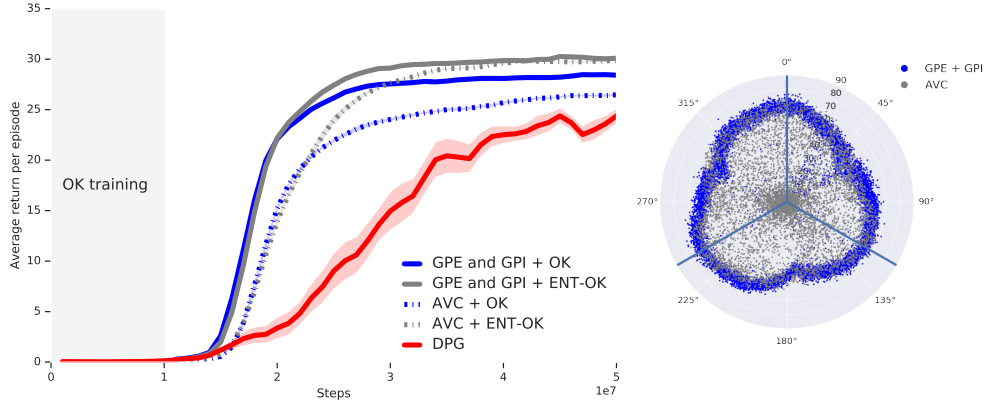

Figure 4: **Left**: Comparison of GPE and GPI with AVC on the moving-target arena. Results were obtained by a DPG player using a standard OK and an entropy-regularised counterpart (ENT-OK). We trained several ENT-OK with different regularisation parameters and picked the one leading to the best AVC performance. The same player and keyboards were used for both methods. Shaded regions are one standard deviation over 10 runs. **Right**: Polar scatter chart showing the average distance travelled by the agent along directions $w$ when combining options using the two competing methods.

is that combining skills in the space of cumulants may be advantageous because it corresponds to manipulating the *goals* underlying the skills. This can be seen if we think of $w \in \mathbb{R}^d$ as a way of encoding skills and compare its effect on behaviour with that of $\theta$: although the option induced by $w_1 + w_2$ through (6) and (7) will seek a combination of both its constituent's goals, the same cannot be said about a skill analogously defined as $\pi(\cdot \mid s; \theta_1 + \theta_2)$. More generally, though, one should expect both policy- and cumulant-based approaches to have advantages and disadvantages.

Interestingly, most of the previous attempts to combine skills in the space of value functions are based on entropy-regularised RL, like the already discussed AVC [34, 9, 11, 13]. Hunt et al. [16] propose a way of combining skills which can in principle lead to optimal performance if one knows in advance the weights of the intended combinations. They also extend GPE and GPI to entropy-regularised RL. Todorov [28] focuses on entropy-regularised RL on linearly solvable MDPs. Todorov [29] and da Silva et al. [6] have shown how, in this scenario, one can compute optimal skills corresponding to linear combinations of other optimal skills—a property later explored by Saxe et al. [20] to propose a hierarchical approach. Along similar lines, Van Niekerk et al. [31] have shown how optimal value function composition can be obtained in entropy-regularised shortest-path problems with deterministic dynamics, with the non-regularised setup as a limiting case.

## 7 Conclusion

The ability to combine skills makes it possible for an RL agent to learn a small set of skills and then use them to generate a potentially very large number of distinct behaviours. A robust way of combining skills is to do so in the space of cumulants, but in order to accomplish this one needs to solve two problems: (1) establish a well-defined mapping between cumulants and skills and (2) define a mechanism to implement the combined skills without having to learn them.

The two main technical contributions of this paper are solutions for these challenging problems. First, we have shown that every deterministic option can be induced by a cumulant defined in an extended domain. This novel theoretical result provides a way of thinking about options whose interest may go beyond the current work. Second, we have described how to use GPE and GPI to synthesise combined options on-the-fly, *with no learning involved*. To the best of our knowledge, this is the only method to do so in general MDPs with performance guarantees for the combined options.

We used the above formalism to introduce OK, an interface to an RL problem in which actions correspond to combined skills. Since OK is compatible with essentially any RL method, it can be readily used to endow our agents with the ability to combine skills. In describing the analogy with a keyboard that inspired our work, Sutton [23] calls for the need of "something larger than actions, but more combinatorial than the conventional notion of options." We believe OK provides exactly that.

## Acknowledgements

We thank Joseph Modayil for first bringing the subgoal keyboard idea to our attention, and also for the subsequent discussions on the subject. We are also grateful to Richard Sutton, Tom Schaul, Daniel Mankowitz, Steven Hansen, and Tuomas Haarnoja for the invaluable conversations that helped us develop our ideas and improve the paper. Finally, we thank the anonymous reviewers for their comments and suggestions.

## Footnotes

[1]A video of the quadrupedal simulated robot being controlled by the DPG player can be found on the following link: https://youtu.be/39Ye8cMyelQ.

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
