[Supplementary Material]

# The Option Keyboard
## Combining Skills in Reinforcement Learning
### Supplementary Material

**André Barreto**, **Diana Borsa**, **Shaobo Hou**, **Gheorghe Comanici**, **Eser Aygün**,
**Philippe Hamel**, **Daniel Toyama**, **Jonathan Hunt**, **Shibl Mourad**, **David Silver**, **Doina Precup**

{andrebarreto,borsa,shaobohou,gcomanici,eser}@google.com
{hamelphi,kenjitoyama,jjhunt,shibl,davidsilver,doinap}@google.com

DeepMind

## Abstract

In this supplement we give details of the theory and experiments that had to be left out of the main paper due to space constraints. We prove our theoretical result, provide a thorough description of the protocol used to carry out our experiments, and present details of the algorithms. We also present additional empirical results and analysis, as well as a more in-depth discussion of several aspects of OK. The numbering of sections, equations, and figures resume from what is used in the main paper, so we refer to these elements as if paper and supplement were a single document.

## A  Theoretical results

**Proposition 1.** *Every extended cumulant induces at least one deterministic option, and every deterministic option can be unambiguously induced by an infinite number of extended cumulants.*

*Proof.* We start by showing that every extended cumulant induces one or more deterministic options. Let $e : \mathcal{H} \times \mathcal{A}^+ \times \mathcal{S} \mapsto \mathbb{R}$ be a cumulant defined in an MDP $M \equiv (\mathcal{S}, \mathcal{A}, p, \cdot, \gamma)$; our strategy will be to define an extended MDP $M^+$ and a corresponding cumulant $\hat{e}$ and show that maximising $\hat{e}$ over $M^+$ corresponds to a deterministic option in the original MDP $M$.

Since we need to model the termination of options, we will start by defining a fictitious absorbing state $s_\emptyset$ and let $\mathcal{H}^+ \equiv \mathcal{H} \cup \{s_\emptyset\}$. Moreover, we will use the following notation for the last state in a history: $\text{last}(h_{t:t+k}) = s_{t+k}$. Define $M^+ \equiv (\mathcal{H}^+, \mathcal{A}^+, \hat{p}, \cdot, \gamma)$ where

$$\hat{p}(has|h, a) = p(s|\text{last}(h), a) \text{ for all } (h, a, s) \in \mathcal{H} \times \mathcal{A} \times \mathcal{S},$$
$$\hat{p}(s_\emptyset|h, \tau) = 1 \text{ for all } h \in \mathcal{H}, \text{ and}$$
$$\hat{p}(s_\emptyset|s_\emptyset, a) = 1 \text{ for all } a \in \mathcal{A}^+.$$

We can now define the cumulant $\hat{e}$ for $M^+$ as follows:

$$\hat{e}(h, a, s) = e(h, a, s) \text{ for all } (h, a, s) \in \mathcal{H} \times \mathcal{A}^+ \times \mathcal{S}, \text{ and}$$
$$\hat{e}(s_\emptyset, a, s_\emptyset) = 0 \text{ for all } a \in \mathcal{A}^+.$$

We know from the dynamic programming theory that maximising $\hat{e}$ over $M^+$ has a unique optimal value function $Q_{\hat{e}}^*$ [19]; we will use $Q_{\hat{e}}^*$ to induce the three components that define an option. First, define the option's policy $\pi_e : \mathcal{H} \to \mathcal{A}$ with $\pi_e(h) \equiv \text{argmax}_{a \neq \tau} Q_{\hat{e}}^*(h, a)$ (with ties broken arbitrarily). Then, define the termination function as

$$\beta_e(h) \equiv \begin{cases} 1 \text{ if } \tau \in \text{argmax}_a Q_{\hat{e}}^*(h, a), \\ 0 \text{ otherwise.} \end{cases}$$

Finally, let $\mathcal{I}_e \equiv \{s \,|\, \beta_e(s) = 0\}$ be the initiation set. It is easy to see that the option $o_e \equiv (\mathcal{I}_e, \pi_e, \beta_e)$ is a deterministic option in the MDP $M$.

We now show that every deterministic option can be unambiguously induced by an infinite number of extended cumulants. Given a deterministic option $o$ specified in $M$ and a negative number $z < 0$, our strategy will be to define an augmented cumulant $e_z : \mathcal{H} \times \mathcal{A}^+ \times \mathcal{S} \mapsto \mathbb{R}$ that will induce option $o$ using the construction above (i.e. from the optimal value function $Q^*_{\hat{e}_z}$ on $M^+$).

First we note a subtle point regarding the execution of options and the interaction between the initiation set and the termination function. Whenever an option $o$ is initiated in state $s \in \mathcal{I}_o$, it first executes $a = \pi_o(s)$ and only checks the termination function in the resulting state. This means that an option $o$ will always be executed for at least one time step. Similarly, an option that cannot be initiated in state $s$ does not need to terminate at this state (that is, it can be that $s \notin \mathcal{I}_o$ and $\beta_o(h) < 1$, with $\mathrm{last}(h) = s$). Given a deterministic option $o \equiv (\mathcal{I}_o, \pi_o, \beta_o)$, let

$$
e_z(h, a, \cdot) = \begin{cases} 0 \text{ if } a = \tau, h \in \mathcal{S} \text{ and } h \notin \mathcal{I}_o, \\ 0 \text{ if } a = \tau, h \notin \mathcal{S} \text{ and } \beta_o(h) = 1, \\ 0 \text{ if } a = \pi_o(h), \text{ and} \\ z \text{ otherwise.} \end{cases}
\tag{9}
$$

We use the same MDP extension $M^+ \equiv (\mathcal{H}^+, \mathcal{A}^+, \hat{p}, \cdot, \gamma)$ as described above and maximise the extended cumulant $\hat{e}_z$. It should be clear that $Q^*_{\hat{e}_z}(h, a) = 0$ only when the action $a$ corresponds to either a transition or a termination dictated by option $o$, and $Q^*_{\hat{e}_z}(h, a) < 0$ otherwise. As such, option $o$ is induced by the set of cumulants $\{e_z \,|\, z < 0\}$ of infinite size. $\qquad\square$

# B  Additional pseudo-code

In this section we present one additional pseudo-code as a complement to the material in the main paper. Algorithm 3 shows a very simple way of building the set $\mathcal{Q}_{\mathcal{E}}$ used by OK through $Q$-learning and $\epsilon$-greedy exploration. The algorithm uses fairly standard RL concepts. Perhaps the only detail worthy of attention is the strategy adopted to explore the environment, which switches between options with a given probability ($\epsilon_1$ in Algorithm 3). This is a simple, if somewhat arbitrary, strategy to collect data, which can probably be improved. It was nevertheless sufficient to generate good results in our experiments.

# C  Details of the experiments

In this section we give details of the experiments that had to be left out of the main paper due to the space limit.

## C.1  Foraging world

### C.1.1  Environment

We start by giving a more detailed description of the environment. The goal in the foraging world is to manage a set of $m$ "nutrients" $i$ by navigating in a grid world and picking up food items containing these nutrients in different proportions. At every time step $t$ the agent has a certain quantity of each nutrient available, $x_{it}$, and the desirability of nutrient $i$ is a function of $x_{it}$, $d_i(x_{it})$. For example, we can have $d_i(x_{it}) = 1$ if $x_{it}$ is within certain bounds and $d_i(x_{it}) = -1$ otherwise. The quantity $x_{it}$ decreases by a fixed amount $l_i$ at each time step, regardless of what the agent does: $x'_{it} = x_{it} - l_i$. The agent can increase $x_{it}$ by picking up one of the many food items available. Each item is of a certain type $j$, which defines how much of each nutrient it provides. We can thus represent a food type as a vector $\boldsymbol{y}_j \in \mathbb{R}^m$ where $y_{ji}$ indicates how much $x_{it}$ increases when the agent consumes an item of that type. If the agent picks up an item of type $j$ at time step $t$ it receives a reward of $r_t = \sum_i y_{ji} d_i(x_{it})$, where $x_{it} = x'_{it-1} + y_{ji}$. If the agent does not pick up any items it gets a reward of zero and $x_{it} = x'_{it-1}$. The environment is implemented as a grid, with agent and food items occupying one cell each, and the usual four directional actions available. Observations are images representing the configuration of the grid plus a vector of nutrients $\boldsymbol{x}_t = [x_{1t}, ..., x_{mt}]$.

The foraging world was implemented as a $12 \times 12$ grid with toroidal dynamics—that is, the grid "wraps around" connecting cells on opposite edges. We used $m = 2$ nutrients and 3 types of food:

---

**Algorithm 3** Compute set $\mathcal{Q}_\mathcal{E}$ with $\epsilon$-greedy $Q$-learning

---

**Require:** $\begin{cases} \mathcal{E} = \{e_1, e_2, ..., e_d\} & \text{cumulants} \\ \epsilon_1 & \text{probability of changing cumulant} \\ \epsilon_2 & \text{exploration parameter} \\ \alpha & \text{learning rate} \\ \gamma & \text{discount rate} \end{cases}$

1: select an initial state $s \in \mathcal{S}$
2: $k \leftarrow \text{Uniform}(\{1, 2, ..., d\})$
3: **repeat**
4:    **if** Bernoulli($\epsilon_1$)=1 **then**
5:       $h \leftarrow s$
6:       $k \leftarrow \text{Uniform}(\{1, 2, ..., d\})$                             // pick a random $e_k$
7:    **if** Bernoulli($\epsilon_2$)=1 **then** $a \leftarrow \text{Uniform}(\mathcal{A})$                // explore
8:    **else** $a \leftarrow \text{argmax}_b \tilde{Q}_{e_k}^{\omega_{e_k}}(h, b)$                        // GPI
9:    **if** $a \neq \tau$ **then**
10:       execute action $a$ and observe $s'$
11:       $h' \leftarrow u(h, a, s')$                            // e.g. $u(h, a, s') = has'$
12:       **for** $i \leftarrow 1, 2, ..., d$ **do**                      // update $Q$-values
13:          $a' \leftarrow \text{argmax}_b \tilde{Q}_{e_i}^{\omega_{e_i}}(h', b)$                // $a' = \omega_{e_i}(h')$
14:          **for** $j \leftarrow 1, 2, ..., d$ **do**
15:             $\delta \leftarrow e_j(h, a, s') + \gamma' \tilde{Q}_{e_j}^{\omega_{e_i}}(h', a') - \tilde{Q}_{e_j}^{\omega_{e_i}}(h, a)$
16:             $\boldsymbol{\theta}_{\omega_i} \leftarrow \boldsymbol{\theta}_{\omega_i} + \alpha \delta \nabla_{\boldsymbol{\theta}_{\omega_i}} \tilde{Q}_{e_j}^{\omega_{e_i}}(h, a)$
17:       $s \leftarrow s'$
18:    **else**                                   // update values associated with termination
19:       **for** $i \leftarrow 1, 2, ..., d$ **do**
20:          **for** $j \leftarrow 1, 2, ..., d$ **do**
21:             $\delta \leftarrow e_j(h, \tau) - \tilde{Q}_{e_j}^{\omega_{e_i}}(h, \tau)$
22:             $\boldsymbol{\theta}_{\omega_i} \leftarrow \boldsymbol{\theta}_{\omega_i} + \alpha \delta \nabla_{\boldsymbol{\theta}_{\omega_i}} \tilde{Q}_{e_j}^{\omega_{e_i}}(h, \tau)$
23: **until** stop criterion is satisfied
24: **return** $\mathcal{Q}_\mathcal{E} \equiv \{\tilde{Q}_{e_j}^{\omega_{e_i}} \mid \forall (i, j) \in \{1, 2, ..., d\}^2\}$

---

$\boldsymbol{y}_1 = (1, 0)$, $\boldsymbol{y}_2 = (0, 1)$, and $\boldsymbol{y}_3 = (1, 1)$. Observations were image-like features of dimensions $12 \times 12 \times 3$, where the last dimension indicates whether there is a food item of a certain type present in a cell. The observations reflect the agent's "egocentric" view, *i.e.*, the agent is always located at the centre of the grid and is thus not explicitly represented. At every step the amount available of each nutrient was decreased by $l_i = 0.05$, for $i = 1, 2$. The desirability functions $d_i(x_i)$ used in the experiments of Section 5.1 were:

$$
\begin{array}{cc}
\textbf{Scenario 1} & \textbf{Scenario 2} \\[4pt]
d_1(x_1) = \begin{cases} +1 & x_1 \leq 10 \\ -1 & x_1 > 10 \end{cases} & d_1(x_1) = \begin{cases} +1 & x_1 \leq 10 \\ -1 & x_1 < 10 \end{cases} \\[12pt]
d_2(x_2) = \begin{cases} -1 & x_2 \leq 5 \\ +5 & 5 < x_2 < 25 \\ -1 & x_2 \geq 25 \end{cases} & d_2(x_2) = \begin{cases} -1 & x_2 \leq 5 \\ +5 & 5 < x_2 < 15 \\ -1 & x_2 \geq 15 \end{cases}
\end{array}
$$

### C.1.2   Agent

**Agents' architecture:** All the agents used a multilayer perceptron (MLP) with the same architecture to compute the value functions. The network had two hidden layers of $64$ and $128$ units with RELU activations. $Q$-learning's network had $|\mathcal{A}| = 4$ output units corresponding to $\tilde{Q}(s, a)$. The network of the $Q$-learning player had $|\mathcal{W}|$ output units corresponding to $\tilde{Q}(s, w)$, while OK's network had $2 \times 2 \times |\mathcal{A}^+|$ outputs corresponding to $\tilde{Q}_{e_j}^{\omega_{e_i}}(h, a) \in \mathcal{Q}_\mathcal{E}$.

(a) Arena                    (b) Quadrupedal simulated robot

Figure 5: The moving-target arena

The states $s$ used by $Q$-learning and the $Q$-learning player were $12 \times 12 \times 3$ images plus a two-dimensional vector $x$ corresponding to the agent's nutrients. The histories $h$ used by OK were $s$ plus an indicator function signalling whether the agent has picked up a food item—that is, the update function $u(h, a, s')$ showing up in Algorithms 1 and 3 was defined as $u(h, a, s') = [\mathbf{1}\{\text{agent has picked up a food item}\}, s']$.

**Agents' training:**   As described in Section 5.1, in order to build OK we defined one cumulant $e_i \in \mathcal{E}$ associated with each nutrient. We now explain in more detail how cumulants were defined. If the agent picks up a food item of type $j$ at time step $t$, $e_i(h_t, a_t, \cdot) = y_{ji}$. After a food item is picked up we have that $e_i(h, a, s) = -\mathbf{1}\{a \neq \tau\}$ for all $h$, $a$, and $s$—that is, the agent gets penalised unless it terminates the option. In all other situations $e_i = 0$.

OK was built using Algorithm 3 with the cumulants $e_i \in \mathcal{E}$, exploration parameters $\epsilon_1 = 0.2$ and $\epsilon_2 = 0.1$, and discount rate $\gamma = 0.99$. The agent interacted with the environment in episodes of length 100. We tried the learning rates $\alpha \in \mathcal{L}_1 \equiv \{10^{-1}, 10^{-2}, 10^{-3}, 10^{-4}\}$ and selected the OK that resulted in the best performance using $w = (1, 1)$ on a scenario with $d_1(x) = d_2(x) = 1$ for all $x$. OK was trained for $5 \times 10^6$ steps, but visual inspection suggests that less than 10% of the training time would lead to the same results.[2]

The $Q$-learning player was trained using Algorithm 2 with the abstract action set $\mathcal{W} \equiv \{-1, 0, 1\}^2 - \{[0, 0]\}$ described in the paper, $\epsilon = 0.1$ and $\gamma = 0.99$. All the agents interacted with the environment in episodes of length 300. For all algorithms (flat $Q$-learning, $Q$-learning + options, and $Q$-learning player) we tried learning rates $\alpha$ in the set $\mathcal{L}_1$ above and picked the configuration that led to the maximum return averaged over the last 100 episodes and 10 runs.

## C.2   Moving-target arena

### C.2.1   Environment

The environment was implemented using the MuJoCo physics engine [30] (see Figure 5). The arena was defined as a bounded region $[-10, 10]^2$ and the targets were circles of radius 0.8. We used a control time step of 0.2. The reward is always 0 except when the agent reaches the target, when it gets a reward of 1. In this case both the agent and the target reappear in random locations in $[-5, 5]^2$.

### C.2.2   Agent

**Agents' architecture:**   The network architecture used for the agents was identical to that used in the experiments with the foraging world (Section C.1). Observations are 29-dimensional with the agent's current $(x, y)$ position and velocity, its orientation matrix ($3 \times 3$), a 2-dimensional vector of distances from the agent to the current target, and two 8-dimensional vectors with angles and

velocities of each joint. The histories $h$ used by OK were simply the length of the trajectory plus the current state, that is, the update function $u(h, a, s')$ showing up in Algorithms 1 and 3 was defined in order to compute $h_{t:t+k} = [k, s_{t+k}]$. As mentioned in the main paper, $\mathcal{A} \equiv [-1, 1]^8$.

**Agents' training:** The set of value functions $\mathcal{Q}_{\mathcal{E}}$ used by OK was built using Algorithm 3 with $Q$-learning replaced by deterministic policy gradient (DPG) [22]. Specifically, for each cumulant $e_i \in \mathcal{E}$ we ran standard DPG and used the same data to evaluate the resulting policies on-line over the set of cumulants $\mathcal{E}$. The cumulants $e_i \in \mathcal{E}$ used were the ones described in Section 5.2, equation (8). During training exploration was achieved by adding zero-mean Gaussian noise with standard deviation $0.1$ to DPG's policy. We used batches of 10 transitions per update, no experience replay, and a target network. The discount rate used was $\gamma = 0.9$. The agent interacted with the environment in episodes of length 300. We swept over learning rates $\alpha \in \mathcal{L}_2 \equiv \{10^{-2}, 10^{-3}, 3 \times 10^{-4}, 10^{-4}, 10^{-5}\}$, and selected the OK that resulted in the best performance in a small set of evaluation vectors $\boldsymbol{w}$ with $w_1, w_2 > 0$ (that is, the directions used for evaluation did not correspond to those of the basic options $\omega_{e_i}$). OK was trained for $10^7$ steps.

The $Q$-learning players were trained using Algorithm 2 with the discrete abstraction action set $\mathcal{W}$ described in the paper, $\epsilon = 0.1$, and $\gamma = 0.99$. Updates were applied to batches of 10 sample transitions. The DPG player was trained using the same implementation as the DPG used to build OK, and the same value $\gamma = 0.99$ used by the $Q$-learning player. Note that, given a fixed $\boldsymbol{w} \in \mathbb{R}^2$, in order to compute the max operator appearing in (7) we need to sample actions $\boldsymbol{a} \in \mathbb{R}^8$. We did so using a simple cross-entropy Monte-Carlo method with 50 samples [35]. The same DPG implementation was also used by the flat DPG as a baseline, the only difference being that it used the actions space $\mathcal{A} \subset \mathbb{R}^8$ instead of the abstract action space $\mathcal{W} \subset \mathbb{R}^2$. All the agents interacted with the environment in episodes of length 1200. For all algorithms ($Q$-learning, $Q$-learning player, DPG, and DPG player) we tried learning rates $\alpha$ in the set $\mathcal{L}_2$ above and picked the configuration that led to the maximum average return, averaged over 10 runs.

The results comparing GPE and GPI with AVC shown in Figure 4 were generated exactly as explained above. In order to train the entropy-regularised OKs we used the soft actor-critic algorithm proposed by Haarnoja et al. [14]. We trained one OK for each regularisation parameter in $\{0.001, 0.01, 0.03, 0.1, 0.3, 1.0\}$ and selected the one leading to the best performance of AVC. In addition to the implementation of the AVC option $\hat{\omega}_e(h)$ described in Section 5.2, we also tried a "soft" version in which $\hat{\omega}_e$ is a stochastic policy defined as $\hat{\omega}_e(a|h) \propto \sum_j w_j \hat{Q}_{e_j}^{\omega_{e_j}}(h, a)$, where, as before, $\hat{Q}_{e_j}^{\omega_{e_j}}(h, a)$ are entropy-regularised value functions and $w_j \in [-1, 1]$. The results with the stochastic policy were slightly worse than the ones shown in Figure 4 (this is consistent with some experiments reported by Haarnoja et al. [14]).

### C.2.3 Experiments

In order to generate the histogram shown in Figure 3 we sampled $100\,000$ directions $\boldsymbol{w} \in \mathbb{R}^2$ from a player with a uniformly random policy and inspected the value of the action $a \in \mathbb{R}^8$ returned by GPI (7). Specifically, we considered the selected action came from one of the three basic options $\omega_{e_i}$ if

$$\min_{i \in \{1,2,3\}, \boldsymbol{a} \in \tilde{\mathcal{A}}} \left| Q_e^{\omega_{e_i}}(h, \omega_{e_i}(h) - Q_e^{\omega_{e_i}}(h, \boldsymbol{a})) \right| \leq 0.15, \tag{10}$$

where $e = \sum_i w_i e_i$ and $\tilde{\mathcal{A}}$ is the set of actions sampled through the cross-entropy sampling process described in the previous section. If (10) was true we considered the action selected came from the option $\omega_{e_i}$ associated with the index $i$ that minimises the left-hand side of (10); otherwise we considered the action came from a combined option.

In order to generate the polar scatter chart shown in Figure 4 we sampled $10\,000$ pairs $(s, \boldsymbol{w})$, with $s$ sampled uniformly at random from $\mathcal{S}$ and abstract actions $\boldsymbol{w}$ sampled from an isotropic Gaussian distribution in $\mathbb{R}^d$ with unit variance (where $d = 3$ for AVC and $d = 2$ for GPE and GPI).[3] Then, for each pair $(s, \boldsymbol{w})$, we ran the option resulting from (6) and (7) for 60 simulated seconds, without

termination, and measured the distance travelled along the desired direction $\boldsymbol{w}$ (for $\boldsymbol{w} \in \mathbb{R}^3$ we first projected the weights onto $\mathbb{R}^2$ using the decomposition discussed in Section 5.2). Each point in the scatter chart defines a vector whose direction is the intended $\boldsymbol{w}$ and whose magnitude is the travelled distance along that direction.

## D    Discussion

In this section we take a closer look at some aspects of OK. We start with a thorough discussion on how extended cumulants can be used to define deterministic options; we then analyse several properties of GPE and GPI in more detail.

### D.1    Defining options through extended cumulants

We have shown that every deterministic option $o$ can be represented by an augmented policy $\omega_e : \mathcal{H} \mapsto \mathcal{A}^+$, which in turn can be induced by an extended cumulant $e : \mathcal{H} \times \mathcal{A}^+ \times \mathcal{S} \mapsto \mathbb{R}$ (in fact, by an infinite number of them). In order to provide some intuition on these relations, in this section we give a few concrete examples of how to generate potentially useful options using extended cumulants.

We start by defining an option that executes a policy $\pi : \mathcal{S} \mapsto \mathcal{A}$ for $k$ time steps and then terminates. This can be accomplished using the following cumulant:

$$e(h, a, \cdot) = \begin{cases} 0 \text{ if length}(h) \leq k \text{ and } a = \pi(\text{last}(h)); \\ 0 \text{ if length}(h) = k + 1 \text{ and } a = \tau; \\ -1 \text{ otherwise,} \end{cases} \tag{11}$$

where $\text{length}(h)$ is the length of history $h$, that is, $\text{length}(h_{t,t+k}) = k + 1$, and $\text{last}(h_{t:t+k}) = s_{t+k}$ (also see (8)). Note that if $\pi(s) = a$ for all $s \in \mathcal{S}$ action $a$ is repeated $k$ times in sequence; for the particular case where $k = 1$ we recover the primitive action $a$. Another instructive example is an option that navigates to a goal state $g \in \mathcal{S}$ and terminates once this state has been reached. We can get such behaviour using the following extended cumulant:

$$e(h, a, \cdot) = \begin{cases} 1 \text{ if last}(h) = g \text{ and } a = \tau; \\ 0 \text{ otherwise.} \end{cases} \tag{12}$$

Note that the cumulant is non-zero only when the agent chooses to terminate in $g$. Yet another possibility is to define a fixed termination bonus $e(h, \tau) = z$ for all $h \in \mathcal{H}$, where $z \in \mathbb{R}$; in this case the option will terminate whenever it is no longer possible to get more than $z$ discounted units of $e$.

Even though working in the space of histories $\mathcal{H}$ is convenient at the conceptual level, in practice the extended cumulants only have to be defined in a small subset of this space, which makes them easy to be implemented. In order to implement (11), for example, one only needs to keep track of the number of steps executed by the option and the last state and action experienced by the agent (*cf.* Section C.2.2). The implementation of (12) is even simpler, requiring only the current state and action. Obviously, one is not restricted to cumulants of these forms; other versions of $e$ can define interesting trade-offs between terminating and continuing.

As a final observation, note that, unlike with standard termination functions $\beta_o(h)$, (5) depends on the value function $Q_e^{\omega_e}$. This means that, when $Q_e^{\omega_e}$ is being *learned*, the termination condition may change during learning. This can be seen as a natural way of incorporating $\beta_o(h)$ into the learning process, and thus impose a form of consistency on the agent's behaviour. When we define (5), we are asking the agent to terminate in $h$ if it cannot get more than $e(h, \tau)$ (discounted) units of $e$; thus, even if it *is* possible to do so, a sub-optimal agent that is not capable of achieving this should perhaps indeed terminate.

### D.2    GPE and GPI

**The nature of GPE and GPI's options**: Given a set of cumulants $\mathcal{E}$, GPE and GPI can be used to compute an approximation of any option induced by a linear combination of the elements of this set. Although this potentially gives rise to a very rich set of behaviours, not all useful combinations of skills can be represented in this way. To illustrate this point, suppose that all cumulants $e \in \mathcal{E}$ take values in $\{0, 1\}$. In this case, when the weights $\boldsymbol{w}$ are nonnegative, it is instructive to think

of GPE and GPI as implementing something in between the `AND` and the `OR` logical operators, as positive cumulants are rewarding in isolation but more so in combination. GPE and GPI cannot implement a strict `AND`, for example, since this would require only rewarding the agent when all cumulants are equal to 1. Van Niekerk et al. [31] present a related discussion in the context of entropy-regularised RL.

**The mechanics of GPE and GPI**: There are two ways in which OK's combined options can provide benefits with respect to an agent that only uses single options. As discussed in Section 3.2, a combined option constructed through GPE and GPI *can be different from all its constituent options*, meaning that the actions selected by the former may not coincide with any of the actions taken by the latter (including termination). But, even when the combined option could in principle be recovered as a sequence of its constituents, having it can be very advantageous for the agent. To see why this is so, it is instructive to think of GPE and GPI in this case as a way of automatically carrying out an alternation of the single options that would otherwise have to be deliberately implemented by the agent. This means that, in order to emulate combined options that are a sequence of single options, a termination should occur at every point where the option achieving the maximum in (7) changes, resulting in potentially many more decisions to be made by the agent.

**Option discovery**: As discussed in Section 3, the precise interface to an RL problem provided by OK is defined by a set of extended cumulants $\mathcal{E}$ plus a set of abstract actions $\mathcal{W}$. A natural question is then how to define $\mathcal{E}$ and $\mathcal{W}$. Although we do not have a definite answer to this question, we argue that these definitions should aim at exploiting a specific structure in the RL problem. Many RL problems allow for a hierarchical decomposition in which decisions are made at different levels of temporal abstraction. For example, as illustrated in Section 5.2, in a navigation task it can be beneficial to separate decisions at the level of intended locomotion (*e.g.*, "go northeast") from their actual implementation (*e.g.*, "apply a certain force to a specific joint"). Most hierarchical RL algorithms exploit this sort of structure in the problem; another type of structure that has received less attention occurs when each hierarchical level can be further decomposed into distinct skills that can then be combined (for example, the action "go northeast" can be decomposed into "go north" and "go east"). In this context, the cumulants in $\mathcal{E}$ should describe the basic skills to be combined and the set $\mathcal{W}$ should identify the combinations of these skills that are useful. Thus, the definition of $\mathcal{E}$ and $\mathcal{W}$ decomposes the problem of option discovery into two well-defined objectives, which can potentially make it more approachable.

**The effects of approximation**: Once $\mathcal{E}$ and $\mathcal{W}$ have been defined one has an interface to a RL problem composed of a set of deterministic options $\tilde{\omega}_e$. Each $\tilde{\omega}_e$ is an *approximation* of the option $\omega_e$ induced by the cumulant $e = \sum_i w_i e_i$. Barreto et al. [3] have shown that it is possible to bound $Q^{\omega_e}_e - Q^{\tilde{\omega}_e}_e$ based on the quality of the approximations $\tilde{Q}^{\omega_{e_i}}_{e_j}$ and the minimum distance between $e$ and the cumulants $e_i \in \mathcal{E}$. Although this is a reassuring result, in the scenario studied here the sub-optimality of the options $\tilde{\omega}_e$ is less of a concern because it can potentially be circumvented by the operation of the player. To see why this is so, note that a navigation option that slightly deviates from the intended direction can be corrected by the other directional options (especially if it is prematurely interrupted, which, as discussed in Section D.1, can be a positive side effect of using (5)). Although it is probably desirable to have good approximations of the options intended in the design of $\mathcal{E}$, the player should be able to do well as long as the set of available options is expressive enough. This suggests that the potential to induce a *diverse* set of options may be an important criterion in the definition of the cumulants $\mathcal{E}$, something previously advocated in the literature.

# E   Additional results and analysis

We now present some additional empirical results that had to be left out of the main paper due to the space limit.

## E.1   Foraging World

In this section we will take a closer look at the results presented in the main paper and study step-by-step the behaviour induced by different desirability profiles of the nutrients. For each of these regimes of desirabily, we will also inquire what the combined options will do and which of them one would expect to be useful. In order to do this, we are going to consider various instantiations of the absract action set $\mathcal{W}$. As a reminder, the $Q$-learning player presented in the main paper

Figure 6: Results on the foraging world using $m = 2$ nutrients and 3 types of food items: $\boldsymbol{y}_1 = (1, 0)$, $\boldsymbol{y}_2 = (0, 1)$, and $\boldsymbol{y}_3 = (1, 1)$. Shaded regions are one standard deviation over 10 runs.

was trained using Algorithm 2 with the abstract action set $\mathcal{W} \equiv \{-1, 0, 1\}^2 - \{[0, 0]\}$. In the following section, we will refer back to this agent as 'Q-learning player (8)', indicating the cardinality of the set of absract action considered – in this case, $|\mathcal{W}| = 8$. In addition, we will consider in our investigations $\mathcal{W}_0 \equiv \{(1, 0), (0, 1)\}$, the set of basic options, and the following individual combinations: $\boldsymbol{w}_1 = (1, 1)$, $\boldsymbol{w}_2 = (1, -1)$, $\boldsymbol{w}_3 = (-1, 1)$, and $\boldsymbol{w}_4 = (-1, -1)$. As in the main paper, we refer to the instantiation of the $Q$-learning player that uses $\mathcal{W}_0$ as *Q-learning player + options* (QO). Otherwise, we will use QP($n$) to refer to a Q-learning player with $n$ (combined) options. Specifically, we adopt QP(3)-$i$ to refer to the players using $\mathcal{W}_i \equiv \mathcal{W}_0 \cup \{\boldsymbol{w}_i\}$. Finally, throughout this study, we include a version of $Q$-learning (QL) that uses the original action space to serve as our flat agent baseline. This agent does not use any form of abstraction and does not have access to the trained options.

Most of the settings of the environment stay the same: we are going to be considering two types of nutrients and three food items $\{\boldsymbol{y}_1 = (1, 0), \boldsymbol{y}_2 = (0, 1), \boldsymbol{y}_3 = (1, 1)\}$ available for pick up. The only thing we are going to be varying is the desirability function associated with each of these nutrients. We will see that this alone already gives rise to very interesting and qualitatively different learning dynamics. In particular, we are going to be looking at four scenarios, slightly simpler than the ones used in the paper, based on the same (pre-trained) keyboard $\mathcal{Q}_\mathcal{E}$. These scenarios should help the reader to build some intuition for what a player based on this keyboard could achieve by combining options under multiple changes in the desirability functions—as exemplified by the scenarios 1 and 2 in the main paper, Figure 2.

The simplest scenario one could consider is one where the desirability functions associated with each nutrients are constant. In particular we will look at the scenario where both desirability functions are positive (Figure 6(c)). In this case we benefit from picking up any of the two nutrients and the most desirable item is item 3 which gives the agent a unit of each nutrient. As our keyboard was trained for cumulants corresponding to $\mathcal{W}_0 = \{(1, 0), (0, 1)\}$, the trained primitive options would be particularly suited for this task, as they are tuned to picking up the food items present in the environment. Performance of the player and comparison with a Q-learning agent are reported in Figure 6(a). The first thing to note is that our player can make effective use of the OK's options and converges very fast to a very good performance. Nevertheless, the Q-learning agent will eventually catch-up and could possibly surpass our players' performance, as our policy for the "true" cumulant induced by $w^* = (1, 1)$ is possibly suboptimal. But this will require a lot more learning.

(a) Scenario $A3$

(b) Scenario $A4$

(c) Scenario $A3$: $d_i(x_i)$

(d) Scenario $A4$: $d_i(x_i)$

Figure 7: Results on the foraging world using $m = 2$ nutrients and 3 types of food items: $\boldsymbol{y}_1 = (1, 0)$, $\boldsymbol{y}_2 = (0, 1)$, and $\boldsymbol{y}_3 = (1, 1)$. Shaded regions are one standard deviation over 10 runs.

The second simple scenario we looked at is the one where one nutrient is desirable—$d_1(x) > 0, \forall x$—and the other one is not: $d_2(x) < 0, \forall x$ (Figure 6(d)). In this case only one of the trained options will be useful, the one going for the nutrient that has a positive weight. But even this is one will be suboptimal as it will pick up equally food item 1 ($\boldsymbol{y}_1 = (1, 0)$) and 3 ($\boldsymbol{y}_3 = (1, 1)$) although the latter will produce no reward for the player. Moreover, sticking to this first option, which is the only sensible policy available to QO, the player cannot avoid items of type 2 if they happen to be on their path to collecting one of the desirable items (1 or 3). This accounts for the suboptimal level that this player, based solely on the trained options, achieves in Figure 6(b). We can also see that the only combined option that improved the performance of the player is $\boldsymbol{w}_3 = (1, -1)$ which corresponds exactly to the underlying reward scheme present in the environment at this time (this is akin to the scenario discussed in Section 3.2, $e_1 - e_2$, where the agent is encouraged to walk ($e_1$) while avoiding grasping objects ($e_2$)). By adding this *synthesised option* to the collection of primitive options, we can see that the player Q(3)-3 achieves a considerably better asymptotic performance and maintains a speedy convergence compared to our baseline (QL). It is also worth noting that, in the absence of any information about the dynamics of the domain, we can opt for a range of diverse combinations, like exemplified by QP(8), and let the player decide which of them is useful. This will mean learning with a larger set of options, which will delay convergence. At the same time this version of the algorithm manages to deal with both of these situations, and many more, as shown in Figures 7 and 8, without any modification, being agnostic to the type of change in reward the player will encounter. We hypothesise that this is representative of the most common scenario in practice, and this is why in the main paper we focus our analysis on this scenario alone. Nevertheless, in this in-depth analysis, we aim to understand what different combination of the same set of options would produce and in which scenarios a player would be able to take advantage of these induced behaviours.

Next we are going to look at a slightly more interesting scenario, where the desirability function changes over time as a function of the player's inventory. An intuitive scenario is captured in $A3$ (Figure 7(c)) where for the second nutrient we will be considering a function that gives us positive reward until the number of units of this nutrient reaches a critical level—in this particular scenario 5—, when the reward associated with it changes sign. The first nutrient remains constant with a positive value. We can think of the second nutrient here as something akin to certain types of food, like "sugar": at some point, if you have too much, it becomes undesirable to consume more. And thus you would have to wait until the leakage rate $l_i$ pushes this nutrient into a positive range before attempting to pick it up again. Conceptually this is a combination of the scenarios we have explored previously

in Figure 6, but now the *two situations would occur in the same episode*. Results are presented in Figure 7(a). As before, we can see that adding the synthesised option corresponding to $\boldsymbol{w}_3 = (1, -1)$, emulating the reward structure in the second part of the episode, gives us a considerable boost in performance as compared to the primitive set $\mathcal{W}_0$ (QO). Moreover, we can see again that the player converges considerably faster than the Q-learning agent which now encounters a switch in the reward structure based on inventory. This change in the desirability function makes this scenario considerably more challenging for the flat agent, while the player has the right level of abstraction to deal with this problem effectively.

The fourth scenario considered in this section is a small modification of the one above, where both nutrients have the "sugar" profile: they both start positive and at one point become negative—see Figure 7(d). We consider different thresholds at which this switch happens for each nutrient, to show that we can deal with asymmetries not present in pre-training. The results are shown in Figure 7(b). Now we can see that this small modification leads to a very different learning profile. The first thing to notice is that the player based on only primitive options, QO, does very poorly in this scenario. This is because this player can only act based on options that pick up items and due to the length of our episodes (300) this player will find itself most of the time in the negative range of these desirability functions. Moreover the player will be unable to escape this range as it will continue to pick up items, resulting in more nutrients being added to its inventory, since these are the only options available to it. On the other hand we can see that by considering combinations of these options our players can do much better. In particular, given the above desirability functions, we expect negative combinations to be helpful. And, indeed, when we add $\boldsymbol{w}_2$, $\boldsymbol{w}_3$ or $\boldsymbol{w}_4$ to the set of primitive options, we can see that the resulting players QP(3)-2,3,4 perform considerably better than QO. Unsurprisingly, adding a positive-only combination, like $\boldsymbol{w}_1$, does not help performance, as even in the positive range this option would be suboptimal and will mimic the performance of the primitive set (as already seen in scenario A1, Figure 6(a)). It is worth noting that in this case we are on par with QL, but keep in mind that this scenario was chosen a bit adversarially against our OK players. Remember this is a scenario where planning on top of the trained options alone would lead to a very poor performance. Nevertheless we have seen that by considering combinations of options, our OK players can achieve a substantially better performance. This is genuinely remarkable and illustrate the power of this method in combining options in cumulant space: *even if the primitive options do not elicit a good plan, combined options synthesised on top of these primitive options can lead to useful behaviour and near optimal performance*.

Lastly, for convenience we included the two scenarios presented in the main paper, as well as their desirability functions, in Figure 8. As in previous analysis, we include the performance of the QP(3)-$i$ players to illustrate how each of these combined options influences the performance. It is worth noting that in all of these scenarios, including the ones in the main text, the same keyboard $\mathcal{Q}_\mathcal{E}$ was (re-)used and that player QP(8), which considers a diverse set of the proposed combinations, can generally recover the best performance. This is an example of an agnostic player that learns to use the most useful combined options in its set, depending on the dynamics it observes in the environment. Moreover, in all of these scenarios we can clearly outperform or at least match the performance of the flat agent, QL, and the agent that only uses basic options, QO. This shows that the proposed hierarchical agent can effectively deal with structural changes in the reward function, by making use of the combined behaviour produced by GPE and GPI (Section 2.1).

### E.2  Moving-target arena

Figure 9 shows additional OK's results on the moving-target arena as we change the length of the options. We vary the options' lengths by changing the value of $k$ in the definition of the cumulants (8). As a reference, we also show the performance of flat DPG in the original continuous action space $\mathcal{A}$.

## Footnotes

[2]Since the point of the experiment was not to make a case in favour of temporal abstraction, we did not deliberately try to minimise the total number of sample transitions used to train the options.

[3]As explained in Section 5.2, in our implementation of GPE and GPI we explored the fact that $Q_{e_{\boldsymbol{w}}}^{\omega}(h, a) = w_1 Q_{e_{\boldsymbol{x}}}^{\omega}(h, a) + w_2 Q_{e_{\boldsymbol{y}}}^{\omega}(h, a)$ for any option $\omega$ and any direction $\boldsymbol{w} \in \mathbb{R}^2$, where $\boldsymbol{x} = [1, 0]$ and $\boldsymbol{y} = [0, 1]$, to only compute two value functions per cumulant $e$. This results in a two-dimensional abstract space $\boldsymbol{w} \in \mathbb{R}^2$. Since GPE is not part of AVC (that is, the option induced by a cumulant is not evaluated under other cumulants), it is not clear how to carry out a similar decomposition in this case.

## References

[35]  P. de Boer, D. Kroese, S. Mannor, and R. Rubinstein,  A Tutorial on the Cross-Entropy Method  *Annals of Operations Research*, 134(1):19–67, 2005.

(a) Scenario 1

(b) Scenario 2

(c) Scenario 1: $d_i(x_i)$

(d) Scenario 2: $d_i(x_i)$

Figure 8: Results on the foraging world using $m = 2$ nutrients and 3 types of items: $\boldsymbol{y}_1 = (1, 0)$, $\boldsymbol{y}_2 = (0, 1)$, and $\boldsymbol{y}_3 = (1, 1)$. Shaded regions are one standard deviation over 10 runs.

(a) $Q$-learning players using $|\mathcal{W}| = 8$ combined options

(b) DPG player

Figure 9: Results on the moving-target arena for options of different lengths. The number of steps corresponds to the value of $k$ in (8). Shaded regions are one standard deviation over 10 runs.