[Reviews · NeurIPS 2019]

Reviewer 1



Post Response update: Thank you for the detailed response. I still believe that a more in depth discussion of the differences or similarities of policy and cumulant based formulations is required to place the paper appropriately in context of prior work. I think the new results presented by the authors in the response partially address my concerns about comparisons with prior work but not fully. I would still like to see comparison against a policy-based method as per the authors' classification. I agree that all methods might have negative transfer but it would be ideal to include a discussion of the conditions under which the methods would show positive or negative transfer (something that the authors do) and to place that in context with other methods at least qualitatively (something that the authors dont). The newer evaluations in the response do satisfy a part of my concerns. I am still not inclined towards a full-fledged accept but do view the paper more favorably. I would recommend the authors to incorporate the changes suggested by the reviewers. Originality: This paper is largely building off of prior research in hierarchical RL and transfer learning. While the methodology is not very distinct from prior work, this paper does bring them together. The core ideas of this paper are shared by many other works transfer learning to guide hiearchical RL ([1],[2],[3]) and parametrizable options [4]. Significance: The paper demonstrates novel ways of combining hierarchical RL and transfer learning, but with so much other work trying to achieve that (see references), but this is not a unique approach towards that, thus it is hard to gauge significance without direct comparisons. Quality and clarity: The paper is technically sound and well structured. I like the idea of using a linear combination of the Q-values (and cumulants) in a generalized policy improvement framework. I also like the clean reformulation of options into generalized policy evaluation and improvement framework through the addition of termination as a primitive action. Major comments: 1) Coverage of options used: As with most approaches that start with limited number of options or have pre-trained options, the quality of the final solution depends very strongly on the options selected for the 'keyboard'. Thus if all the component actions are poor at the 'test' task, the resulting policy would also have no hope of performing well at it. 2) Distinction from entropy-regularized RL: The author's approach might be seen as a special case of entropy regularized RL as T -> 0 [1]. 3) Benefits/drawbacks w.r.t other continuous parametrizations of options: The authors have provided a single way of parametrizing options through linear combination of their Q functions followed by generalized policy evaluation and improvement. However [3] and [4] provide alternative approaches to parametrization of options. How do these methods compare to yours? 4) Clarity on the cumulant used in Foraging domain: The cumulant function used for each of the nutrients as defined in the paper and the supplementary material differ (line 246 in the paper) and (502) in the supplementary. I would request the author to clarify which one it was, and whether the difference was only in the learning curve for the constituent options or was there a difference in the performance of the options keyboard as well. Minor comments: 1) Behavior of Q-learning with simple options: I am curious as to why simple options showed no learning effects in both the scenarios? [1] - Haarnoja, Tuomas, et al. "Composable deep reinforcement learning for robotic manipulation." 2018 IEEE International Conference on Robotics and Automation (ICRA). IEEE, 2018. [2] - Frans, K., Ho, J., Chen, X., Abbeel, P., & Schulman, J. (2017). Meta learning shared hierarchies. arXiv preprint arXiv:1710.09767. [3] - Gupta, Abhishek, et al. "Meta-reinforcement learning of structured exploration strategies." Advances in Neural Information Processing Systems. 2018. [4] - Da Silva, B., Konidaris, G., & Barto, A. (2012). Learning parameterized skills. arXiv preprint arXiv:1206.6398.

Reviewer 2



Post rebuttal update: I thank the authors for taking the time to prepare a detailed response to my questions, suggestions, and criticisms. I believe that they did a good job in addressing my main questions and concerns. I do agree with other reviewers that comparisons with existing (policy-based) techniques could make this a stronger submission. However, I also agree with the authors that both policy- and cumulant-based approaches should have advantages and disadvantages, so it make sense for them to co-exist in the literature. I suggest trying to better situate the proposed method within the existing literature and to discuss possible downsides of this general approach---e.g. the fact that the simple linear combination of cumulants may often be insufficient to properly span the space of desired behaviors. Given the concerns of other reviewers, I have lowered my score from a 9 to an 8. Overall, though, I still believe that this is an important contribution to the field. ---- This paper introduces a method to combine existing options not merging or analyzing them in the space of policies or actions, but by manipulating them in the space of pseudo-rewards, or cumulants. In particular, the authors first show that every deterministic option can be represented as a cumulant function defined over an extended domain that includes an option-termination action; then, they show that novel options can be obtained by linearly combining the cumulants of the existing options. Importantly, the authors also show that if the cumulants for a given set of existing options is known a priori, the policy for the novel option being synthesized can be directly identified without requiring any learning. This is an interesting paper introducing a novel and important contribution to the field. It is very well written and the experiments clearly demonstrate the contribution of the proposed method. I have few comments and questions: 1) I understand that it may be possible to synthesize novel options by linearly combining their corresponding cumulants. However, I believe there must exist some implicit assumption on what type of behaviors that can be practically defined by computing linear combinations of their corresponding pseudo-reward functions. E.g., the authors introduced, as a motivating example, the goal of learning to walk while grasping an object, given separate options for walking and for grasping. Would it be possible, via the proposed technique, to synthesize options whose constituent options may interfere with one another? The idea of linearly merging reward functions seems to ignore the fact that, in some cases, a reward function for achieving a specific combined goal may need to be more sophisticated than a simple linear scaling of how desirable each of the two (possibly conflicting) sub-goals are. Could you please discuss this possible issue (and the possible underlying assumptions of this approach) in more details? 2) in Eq2, what is "t"? This time variable does not appear anywhere in the definition of the quantity being defined---Q^pi_c(s,a); 3) immediately before Eq3, how is Q^max_c computed, exactly? The text says that Q^max_c = max_i Q^{pi_i}_c, so I assume it corresponds to a maximization over each possible policy pi_i. However, that seems to assume that one Q-function Q1 would always be strictly larger than another Q-function Q2 (i.e., so that Q1(s,a)>Q2(s,a) for all s and a). Shouldn't this max operation, however, be defined over the individual Q-values for each (s,a) under each policy pi_i? 4) the discussion (in page 4, in the paragraph immediately after Eq5) for how the initiation set I_e is defined is not immediately clear to me. It seems to assume that the initiation set is formed by *all* states in which the option does not directly terminate; but this shouldn't necessarily be the case: the termination set may be defined as a strict subset of those states. Could you please clarify or justify this statement/definition? 5) the proposed method seems to assume that, in order to combine options, one needs to know (in advance) the Q-function of each option's policy when evaluated under the cumulant function of all other possible options. Is this correct? If so, how scalable is the method, in the sense that it may require each possible behavior to be evaluated a priori under every other possible pseudo-reward function? 6) in the experiments section (section 5.1) the authors say that "We used Algorithm 3 to compute 4 value functions in Q_eps". It is not immediately clear to me why there are 4 value functions in this case, since m=2 and there are 3 types of good. Could you please clarify? 7) it is very hard to read Fig3 in a black-and-white printed version of the paper. I suggest picking a better set of colors for the curves, if possible.

Reviewer 3



This paper presents a hierarchical reinforcement learning (HRL) framework. The proposed HRL is based on the option framework where the option policies are trained with generalized policy improvement (GPI). Based on the property of GPI, a new option can be obtained by combining evaluated options. The higher-level policy that activates the option is learned through a variant of Q-learning. Originality: Composition of policies used in this paper is briefly mentioned in the original paper by Barreto et al. [2017]. In addition, other methods for composing policy is investigated in [Hunt et al., ICML2019] and [Haarnoja, ICML2018]. Thus, I think the originality of this paper is minor. Quality: The proposed method is a simple extension of GPI to the option framework, and theoretical contribution is minor. Although the empirical results show the benefit of the composing option policies, the performance of the proposed method is not compared with existing methods. Significance: The paper presents a method for obtaining a new option by combining learned options. Although this feature may be novel in the option framework, I think that this contribution is incremental since composing a new policy from existing policies have been investigated in recent studies. === comments after the rebuttal === I appreciate the authors' response. I would like to request authors to address the following points when revising the paper. - What is reported in the rebuttal is comparison of the higher-level policy for composing a new option. The result does not mean that the proposed framework outperforms the method in [ Haarnoja et al., 2018] because the options are not trained with soft-Q learning. Authors should explicitly state this point when addressing the new results to the paper. - I also think that the fact that the options need to be pre-trained is a big limitation since many "policy-based" HRL methods can jointly train the higher- and lower-level policies. This property can be critical in performance since the composition would work very well if the option cumulants properly represents the task dynamics, and if not, it works very poorly as pointed out by other reviewers. In the rebuttal, it is mentioned that "OK and player can be learned together (we are currently working on it).", which means that it has not been achieved yet. I would like to ask authors to honestly discuss the difficulty of jointly training OK and players in their framework.

[Author Response · NeurIPS 2019]

Thank you for the thoughtful reviews! The main concern seems to be the need for a more thorough contextualisation of
OK within the related literature, so we start by addressing this point. Many transfer methods build on the following idea:
first learn a parametric representation of a policy $\pi(\cdot|s; \boldsymbol{\theta})$ that captures the structure of a set of tasks, then quickly adapt
to a new task by fine-tuning $\boldsymbol{\theta}$ (Da Silva *et al.*, 2012; Gupta *et al.*, 2018) or by learning a policy that uses $\boldsymbol{\theta}$ as actions
(Frans *et al.*, 2017; Haarnoja *et al.*, ICML, 2018). We call these *policy-based* methods. One of the central arguments in
our paper is that working in the space of *cumulants* (rather than policies) may offer some advantages: it is a robust
approach because it captures the *intentions* behind the skills being transferred (lines 38–42) and it can generate options
that are not in the policy space "spanned" by their constituents (lines 165–167). Both policy- and cumulant-based
approaches should have advantages and disadvantages, so it is desirable that they co-exist in the literature.

All the papers cited by the reviewers describe policy-based transfer
methods, with 3 exceptions: [1,2,3] describe approaches to compose
policies based on their *value function*. Although [1,2,3] are competi-
tors among themselves, they are *not* direct competitors of OK. OK
extends [1] from policies to options in order to get temporal abstrac-
tion (the benefits of which are well known). Dealing with termination
and initiation of options in a principled way is not a trivial extension.
This involved 3 steps: (i) augment the definition of cumulants to
depend on histories and to also include a termination action, (ii) show
the mapping between the resulting extended cumulants and options
(Prop. 1), and (iii) adapt the machinery in [1] to this more general
scenario. To the best of our knowledge, the resulting OK framework
is *the only way to combine options in the space of cumulants with*
*performance guarantees for general MDPs*. Although OK is not ad-
dressing the same problem as [1,2,3], it is reasonable to ask whether
(iii) could also be applied to [2,3], as suggested by R1 and R3. To
answer this question we implemented a version of OK that uses [2]

Figure 1: OK using [1] and [2]. Curves are the best
result over a set of learning rates and, for [2], also
over temperature parameters for the softmax policy.

rather than [1] to combine options. Specifically, we replaced GPE (eq. 6) and GPI (eq. 7) with the composition of value
functions from [2]: $\tilde{\omega}_e(h) \in \arg\max_{a \in \mathcal{A}^+} \sum_j Q_{e_j}^{\omega_{e_j}}(h, a)$. We also implemented a "softmax" version in which $\tilde{\omega}_e(h)$
is computed using eq 2 of [2]. The results in Fig. 1 suggest that GPE and GPI are more effective than [2] in this case.
We'll add a more extensive version of this comparison to the paper and also a comparison with the two methods in [3].

**R1** MAJOR COMMENTS. **(1)** We are not addressing the problem of option discovery, but we believe that the formalism
we developed allows for a clean formulation of the problem in the space of cumulants (lines 584–598). **(2)** The
composition of value functions proposed in [2] is inherently different from GPE and GPI because an option is never
evaluated under another option's cumulant: since there is no GPE, composition is made with $Q_{e_j}^{\omega_{e_j}}(h, a)$ rather than
with $Q_e^{\omega_{e_j}}(h, a)$ (compare the above with eqs. 6 and 7). **(3)** These are policy-based transfer methods as defined above,
with the associated advantages and disadvantages. **(4)** The information in the appendix is outdated, thank you for
pointing that out! After a food item is consumed, we can either reward the termination $\tau$ or penalize actions $a \neq \tau$.
Although the resulting options are identical, the latter scheme leads to faster learning, and thus it was used for all the
experiments. MINOR COMMENTS. **(1)** Q-learning uses only 2 options. This results in fast learning whose curve looks
flat at the scale of the other methods' curves. But $Q$-learning's non-trivial performance shows that it does learn the task.

**R2 (1)** Some combinations of options are indeed not representable as linear combinations of cumulants. When the
weights $w$ are nonnegative, it is instructive to think of GPE and GPI as something in between an AND and an OR (as
cumulants are rewarding in isolation but more so in combination). GPE and GPI cannot implement a strict AND, for
example. **(2)** $t$ is implicit in the definition of $\mathbb{E}_{s,a}^{\pi}[\cdot]$ (line 73). **(3)** You are correct: the max operator is applied to each
$(s, a)$ independently. We will clarify. **(4)** Suppose that $\mathcal{I}_e = \mathcal{S}$. Since in the states $s$ where $\beta_e(s) = 1$ (*cf.* eq. 5)
executing option $o_e$ will have no effect, we simply exclude those from $\mathcal{I}_e$. This allows us to have $o_e$ be fully determined
by $e$, without any extra definitions. **(5)** If you think of the set $\mathcal{Q}_{\mathcal{E}}$ as a *cumulants* $\times$ *options* matrix, it is possible to
disassociate these quantities. It is true nevertheless that each option must be evaluated under the cumulants we want
to generalize over. The premise is that with a small number of both we can create a very diverse set of behaviours.
We'll elaborate in the appendix. **(6)** We had 2 cumulants associated with goods and 1 policy induced by each cumulant,
resulting in 2 policies $\times$ 2 cumulants = 4 value functions. **(7)** We will add line patterns as we did in Fig. 1, thanks!

**R3** Our main theoretical result, Prop. 1, is largely independent of [1], and we believe its interest goes beyond the
scope of this paper. • The options were learned before (see gray area in Fig. 3 and discussion in lines 508–509 of the
appendix), but in principle OK and player can be learned together (we are currently working on it). • We kindly ask the
reviewer to reconsider their assessment of the significance of the paper in light of the explanations above.

REFERENCES. **[1]** Barreto *et al.* NIPS, 2017. **[2]** Haarnoja *et al.* IEEE ICRA, 2018. **[3]** Hunt *et al.* ICML, 2019.


[Meta-Review · NeurIPS 2019]

The reviewers unanimously recommend accept.